# Communication-Efficient Desire Alignment for Embodied Agent-Human Adaptation

## Abstract

While embodied agents have made significant progress in performing complex physical tasks, real-world applications demand more than pure task execution. The agents must collaborate with unfamiliar agents and human users, whose goals are often vague and implicit. In such settings, interpreting ambiguous instructions and uncovering underlying desires is essential for effective assistance. Therefore, fast and accurate desire alignment becomes a critical capability for embodied agents. In this work, we first develop a home assistance simulation environment **HA-Desire** that integrates an LLM-driven proxy human user exhibiting realistic value-driven goal selection and communication. The ego agent must interact with this proxy user to infer and adapt to the user's latent desires. To achieve this, we present a novel framework **FAMER** for fast desire alignment, which introduces a desire-based mental reasoning mechanism to identify user intent and filter desire-irrelevant actions. We further design a reflection-based communication module that reduces redundant inquiries, and incorporate goal-relevant information extraction with memory persistence to improve information reuse and reduce unnecessary exploration. Extensive experiments demonstrate that our framework significantly enhances both task execution and communication efficiency, enabling embodied agents to quickly adapt to user-specific desires in complex embodied environments.

## 1 Introduction

Embodied AI has seen rapid progress in recent years, driven by the collection of large datasets Brohan et al. (2022; 2023); O'Neill et al. (2024); Fang et al. (2023); Wang et al. (2025b) and the development of large vision-language-action (VLA) models Black et al. (2024); Driess et al. (2023); Kim et al. (2024); Team et al. (2024). These advances have paved the way for general-purpose robots capable of performing complex manipulation tasks in the physical world. However, real-world deployment of embodied agents requires more than physical capabilities. It also demands the ability to interact effectively with diverse human users.

One of the key challenges in such interactions lies in the variability of human preferences, values, and behaviors. Unlike physical tasks with well-defined goals, human desires are often ambiguous, context-dependent, and implicit. For an embodied agent to be truly helpful, it must be able to rapidly infer and align with the user's underlying desires even when explicit instructions are vague or lacking.

A prime example is the home assistant robots. Even if these robots are trained on broad human-centric datasets, they inevitably face unfamiliar users whose specific values and preferences are unknown at deployment. To offer effective assistance, the agent must infer and adapt to these user-specific attributes. In this way, the agent minimizes repetitive communication and demonstrates proactive, personalized behavior, similar to a considerate human assistant. For instance, as illustrated in Figure 1, a robot enters a new home without prior knowledge of the user. Over time, it gradually learns that the user is allergic to caffeine and prefers something refreshing for breakfast due to poor sleep caused by a heavy workload. Therefore, the robot infers that the user wants juice and serves it without needing to ask. Such behavior highlights the necessity of rapid and accurate desire alignment in embodied assistance, enabling robots to build trust and deliver truly helpful service.

Previous works have investigated collaboration with unfamiliar partners under the paradigms of ad-hoc teamwork (AHT) Rahman et al. (2021); Ravula (2019); Stone et al. (2010) and zero-shot coordination (ZSC) Cui et al. (2021); Hu et al. (2021; 2020). However, these efforts have primarily

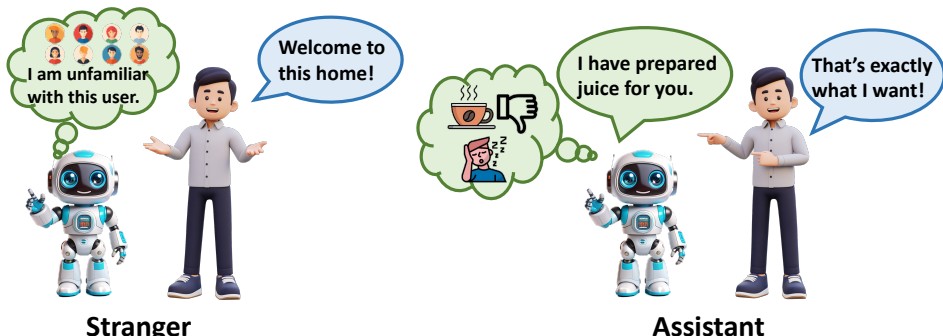

Figure 1: An illustration of Embodied Agent-Human Adaptation. The embodied home assistant robot encounters a new human user with unknown values and preferences. Through interaction over time, the agent learns the user's aversion to caffeine and preference for refreshing drinks in the morning due to inadequate sleep. By aligning with the user's implicit desires, the agent proactively serves juice without being explicitly instructed, demonstrating high-quality assistant service.

focused on simplified domains such as board games like Hanabi Bard et al. (2020) and 2D grid-based simulations like Level-Based Foraging Albrecht & Ramamoorthy (2015) and Overcooked Carroll et al. (2019). These environments lack human-like, value-driven goal specification, natural communication and embodied actions, limiting their applicability to realistic embodied agent-human adaptation.

To bridge this gap, we propose **HA-Desire** (Home Assistance with diverse Desire), a new embodied simulation environment built on VirtualHome Puig et al. (2018) that offers rich 3D household scenes, diverse objects, and tasks such as preparing an afternoon snack. Crucially, it includes an LLM-driven proxy human user that samples goals from assigned value attributes. Real human users may provide imprecise instructions, either because their goals are still being formed or because they wish to minimize communication effort. To reflect this setting, the proxy user communicates with the agent in natural language without explicitly revealing its goals, instead offering indirect hints such as "I want something sweet and crunchy." This design forces the agent to perform strategic inference and interactive reasoning to uncover latent desires and complete tasks that satisfy the user.

To tackle this challenging problem of fast agent-human adaptation, we propose **FAMER** (**F**ast **A**daptation via **ME**ntal **R**easoning), a novel framework that leverages the reasoning capabilities and commonsense knowledge of large vision-language models to improve both communication efficiency and task execution. At the core of FAMER is a desire-centered mental reasoning module that extracts confirmed goals from user messages and infers the user's underlying mental state, including values, preferences, and latent desires. To reduce redundant communications, FAMER also incorporates a reflection-based communication mechanism that prompts the agent to reason about what has already been inferred and to ask only for missing or unconfirmed information. Additionally, FAMER includes a goal-relevant information extraction module that identifies critical task-related details, such as object containers or room locations, and stores them in a persistent memory across episodes. This enables the agent to reuse previously gathered information and avoid unnecessary exploration. Together, these components allow the ego agent to rapidly align with the user's desires and plan efficiently in complex, multi-step embodied tasks.

We evaluate FAMER on two representative tasks: Snack & Table, in our HA-Desire environment. Each task is tested under two settings: Medium and Large, denoting the number of goals to satisfy. Extensive experiments with the LLM-driven proxy user and real human users show that FAMER significantly outperforms baselines in task completion score and communication efficiency. Ablation studies further highlight the contribution of each key component in the framework.

In summary, our contributions include:

- We formulate the novel problem of rapid adaptation to value-driven, unknown users in embodied settings, and introduce HA-Desire, a 3D simulation environment featuring naturalistic user interactions for evaluating agent-human adaptation.

- We propose FAMER, a new framework that integrates desire-centered mental state reasoning, reflection-based efficient communication, and goal-related key information extraction to enable fast desire alignment for embodied agents.

- We demonstrate the effectiveness of our proposed environment and framework through extensive quantitative and qualitative experiments.

## 2 RELATED WORK

**Value Alignment** has been extensively studied in both language models and agent design. In the context of LLMs, alignment techniques such as RLHF Ouyang et al. (2022); Dai et al. (2023); Ji et al. (2023) aim to align models with human preferences, but these efforts primarily focus on static, text-based tasks and do not address the challenges of dynamic, embodied interactions. In human-AI collaboration, value alignment involves inferring user preferences through feedback Yuan et al. (2022); Hiatt et al. (2017); Fisac et al. (2020). More closely related to our setting are mental reasoning agents inspired by Theory of Mind Rabinowitz et al. (2018); Wang et al. (2022), which model other agents' beliefs and desires to support assistance. D2A Wang et al. (2025a) simulates human desires using LLMs, but is limited to text-based tasks. CHAIC Du et al. (2024) introduces an embodied social intelligence challenge that focuses on reasoning under physical constraints, but does not address the diversity of human values and goals. In contrast, our work introduces an embodied simulation platform with naturalistic, value-driven goal generation and communication.

**Adaptive Agents.** Adaptation in multi-agent settings has been studied under the paradigms of zero-shot coordination (ZSC) Hu et al. (2020; 2021); Cui et al. (2021); Lupu et al. (2021); Strouse et al. (2021) and ad-hoc teamwork (AHT) Stone et al. (2010); Rahman et al. (2021); Chen et al. (2020); Mirsky et al. (2020); Ma et al. (2024), where agents must coordinate with unseen partners without prior agreement. While these approaches are effective in structured domains such as Hanabi Bard et al. (2020) and Overcooked Carroll et al. (2019), they rely on symbolic observations, making them less suitable for complex, embodied human-agent collaboration. More recently, LLM-based agent frameworks Li et al. (2023); Yao et al. (2023); Zhang et al. (2024b;a); Liu et al. (2025) have demonstrated impressive capabilities in reasoning and planning within interactive environments. However, most assume known goals or static user preferences and lack mechanisms for inferring latent values through interaction. Our work builds on this line by addressing the challenge of fast adaptation to unknown, value-driven user goals via desire inference, memory utilization, and efficient, human-like dialogue in rich embodied tasks.

## 3 EMBODIED HOME ASSISTANCE SIMULATION ENVIRONMENT

In real-world interactions, human users may provide vague instructions because their goals have only been partially formed and expressed at an abstract level Bettman et al. (1998). At the same time, people generally dislike reiterating their needs during communication Clark & Brennan (1991). Consequently, an effective assistant agent must be able to infer the latent goals and preferences of the human users and adapt its behavior accordingly, thereby minimizing redundant communication and execution costs to provide proactive assistance.

To study these challenges of agent–human adaptation in an embodied environment, we present **HA-Desire** (Home Assistance with diverse Desire), a novel embodied simulation environment designed to study agent-human adaptation in realistic home scenarios. As illustrated in Figure 2, HA-Desire builds upon VirtualHome Puig et al. (2018) and extends it with value-driven proxy human users. It includes 6 distinct home layouts containing typical rooms such as living rooms, kitchens, bathrooms, and bedrooms, with an average of 80 objects per room drawn from over 110 object classes. This diversity of layouts and object assets enables the construction of visually grounded tasks with high variability, providing a realistic testbed for embodied agent-human adaptation.

We instantiate a proxy human user within the environment. This user is assigned a set of latent value attributes and begins with only a vague task description. Guided by these values, the user samples a subset of desire-related goals from a larger goal space. Crucially, these goals are not explicitly revealed; instead, the proxy user provides only indirect hints about preferences and intentions in response to the ego agent's inquiries.

We describe the problem formulation and proxy human user model in more detail below.

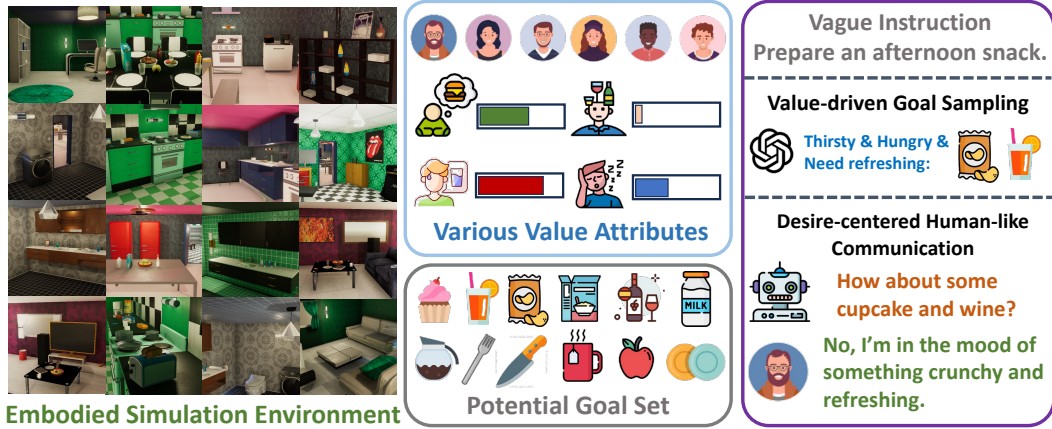

Figure 2: Overview of the HA-Desire environment. The simulation environment contains diverse objects and scenes. The proxy human user samples value attributes from a task-related space, which guides goal selection from a potential goal set via LLM. The user is constrained from directly revealing the true goals and instead communicates through desire-centered hints. This setup encourages the ego agent to infer user intent through interactive reasoning rather than relying on explicit instructions.

## 3.1 PROBLEM FORMULATION

In the desire-centered agent-human adaptation problem, the task goal is not explicitly fixed, but the number of goals is predefined. Instead, there exists a potential goal set $G_p$, from which the task goal must be sampled by the human user $H$ and inferred by the ego agent $E$. Given a vague task description $T$, the proxy human user first samples a set of value attributes $V$ from a task-specific value space. Based on these values and the task description, the user then samples a set of desire-driven goals $G = G(V, G_p, T) \subset G_p$.

The true goal set $G$ is latent and not directly observable by the ego agent, which must infer it through interaction. The ego agent and the human user can communicate by exchanging messages $M_E$ and $M_H$, respectively. The ego agent performs actions according to its policy $\pi(A|O, M_H, C, T)$, where $O$ denotes the current observation of the environment and $C$ is a cross-episode memory context, including previous actions and dialogue history.

During task execution, the ego agent receives a positive reward for successfully completing a true sub-goal in $G$, and a penalty for executing irrelevant or incorrect goals, which reflects a misalignment with the user's desires. The environment supports multi-episode interactions, where the agent repeatedly engages with the same user. The user's value attributes $V$ are consistent across episodes, encouraging the agent to gradually build an internal model of the user.

The objective of the ego agent is to maximize cumulative reward by accurately inferring the user's latent desires, while minimizing interaction steps and communication costs. This promotes both task and communication efficiency, which are critical for effective real-world embodied assistance.

## 3.2 VALUE-DRIVEN HUMAN USER

As illustrated in Figure 2, the proxy human user begins by sampling discretized value attributes $V$ from a task-related value space. For example, in the Prepare Snack task, the value space spans five dimensions: Hungry, Thirsty, SweetTooth, Fruitarian, and Alcoholic, each taking on one of three discrete levels—Not, Somewhat, or Very. These attributes reflect the user's latent desires.

Once the value attributes are sampled, the user invokes a large language model to simulate realistic goal selection. Conditioned on the vague task description $T$ and the sampled values $V$, the LLM generates a set of corresponding desire-related goals $G = G(V, G_p, T) \subset G_p$. Since multiple goals may align with the same value attribute, this sampling process is intentionally non-deterministic, mirroring the variability and ambiguity of human decision-making.

To simulate natural and indirect human communication, the user is constrained via LLM prompting and output filtering to ensure that the true goal set is never revealed explicitly. Instead, it responds

Figure 3: Overview of the FAMER framework. FAMER comprises three key components: KeyInfo Extraction, Desire-Centered Mental Reasoning (including Goal Confirmation and Desire Inference), and Efficient Communication. These are supported by a memory module, perception module, and planning module, which together form an integrated pipeline for embodied agent-human adaptation.

with value-driven hints that reflect its preferences and desires. This encourages the ego agent to reason about the user's intent through interaction to provide proactive help with minimal repetitive annoying confirmation. The detailed LLM prompting strategy used to sample goals and generate user responses is provided in Appendix B.

## 4 COMMUNICATION-EFFICIENT AGENT-HUMAN ADAPTATION

To enable fast and communication-efficient adaptation to users, we propose a novel framework, **FAMER** (Fast Adaptation via MEntal Reasoning), which leverages the reasoning capabilities and commonsense embedded in large VLMs. In this work, we utilize GPT-4o Hurst et al. (2024).

As illustrated in Figure 3, FAMER integrates three core components: Key Information Extraction, Desire-Centered Mental Reasoning, and Efficient Communication. These modules work in concert to help the ego agent infer user intent, plan accurately, and minimize redundant interactions and communication. We detail each component in the following subsections.

### 4.1 INFORMATION EXTRACTION

In HA-Desire, the ego agent receives first-person RGB-D images as observations. To extract structured information from these inputs, we employ a perception module based on Mask R-CNN He et al. (2017), trained on collected scene images following Zhang et al. (2024b). The module first predicts instance segmentation masks from the RGB image and then constructs 3D point clouds using the RGB-D data. These outputs are used to build a scene graph that encodes object locations, categories, and spatial relationships.

We then introduce the first core component of FAMER: Key Information Extraction. With the confirmed and inferred goals context, this module filters and stores goal-relevant information extracted from the scene graph into a dedicated cross-episode memory buffer. For example, if the agent identifies that juice is located in the fridge within the kitchen, it stores the structured entry "juice in fridge in kitchen" in memory. During subsequent planning phases, this stored information is used to guide attention to known facts and reduce redundant exploration. As a result, the agent is better equipped to reuse previously acquired knowledge across episodes, improving task efficiency.

### 4.2 DESIRE-CENTERED MENTAL REASONING

This module is composed of two interconnected components: Goal Confirmation and Desire Inference, as illustrated in Figure 3. Together, they enable the agent to infer the user's underlying desires.

The Goal Confirmation component extracts confirmed goals from the user's responses by VLM reasoning. For example, if the agent asks, "Do you want some juice?" and the user replies, "Correct!

Try to look for something crunchy," the system confirms that juice is one of the user's desired items. This process grounds part of the goal set and reduces future uncertainty.

Following confirmation, the Desire Inference component leverages the action & dialogue history, confirmed goals and past episode goals to reason about the user's mental state, including value attributes and desires. Since user values remain consistent across episodes, the agent can incrementally improve its inference accuracy over time. By maintaining an internal model of the user, the agent can focus on narrowing down the remaining potential goals and avoid repetitive or redundant guesses.

With both inferred and confirmed goals in hand, the agent filters out irrelevant actions during planning. As shown in Figure 3, if the current goals do not involve a toothbrush or candle, then actions involving those objects are ignored. This reduces distraction and helps the agent maintain focus on goal-relevant objects and activities, thereby improving planning efficiency and task performance.

### 4.3 Efficient Communication

Excessive or repetitive communication can diminish user satisfaction and hinder overall system efficiency. To address this, FAMER integrates an Efficient Communication module that promotes purposeful, context-aware dialogue between the agent and the user.

This module leverages both the dialogue history and the agent's inferred model of the user's mental state to decide when and what to ask. Before initiating a new query, the agent performs an internal reflection over its current knowledge—what goals have been confirmed, which value attributes have been inferred, and what uncertainties remain. This reflective mechanism helps avoid redundant or previously resolved questions, particularly across multi-episode interactions.

When communication is necessary, the agent formulates targeted, desire-aligned questions aimed at resolving specific ambiguities. For example, if the agent has inferred a preference for sweet items but is uncertain about texture, it may ask "I found a cupcake. Would you like it?" instead of issuing vague or open-ended queries. This focused interaction strategy minimizes the communication burden on the user while enabling the agent to efficiently acquire high-value information.

## 5 Experiment

### 5.1 Tasks & Metrics

We evaluate FAMER in two representative tasks instantiated within our proposed HA-Desire environment: **Prepare Afternoon Snack** and **Set Up Dinner Table**. Each task is associated with a five-dimensional value space that governs user preferences. The Snack task includes 10 potential goals, while the Table task contains 8. Each task is further divided into two levels: Medium and Large. In Medium setting, the number of target goals is 2, and the maximum episode length is 60 steps. In Large setting, the agent must satisfy 4 goals within 120 steps. For example, the Snack-M task involves a total of $C(10, 2) = 45$ possible goal combinations. The reason why we set the maximum goal space to 4 varying objects is that for daily tasks such as meal preparation, people often share a stable backbone of common objects and a small consideration subset of items ($\approx$3–5) that vary depending on individual preferences due to limited cognitive constraints Schank & Abelson (1975); Wood et al. (2022); Hauser & Wernerfelt (1990). Details of the tasks are provided in Appendix B.

For each task at a given level, every method is evaluated over 6 independent runs. In each run, the agent interacts with the human user for 3 episodes, with the user's value attributes fixed throughout to match the agent-human adaptation setting. All reported results are averaged over the 6 runs.

We evaluate performance using the following metrics:

**Score**: Given $N$ total goals, the agent receives a reward of $\frac{1}{N}$ for each correct goal achieved. Completing an incorrect or distracting goal incurs a penalty of $-\frac{1}{2N}$. The maximum achievable score per episode is 1, corresponding to the completion of all goals without any mistakes.

**Communication Cost**: The total number of tokens exchanged in messages between the agent and the user, including both queries and responses.

Notably, HA-Desire supports diverse objects and scenes, and goal generation is automated, making it easy to extend to other tasks. Further details of the environments are provided in Appendix B.

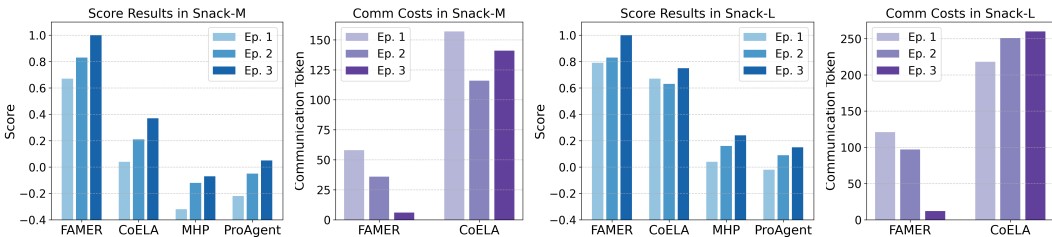

Figure 4: Quantitative results in Snack-M and Snack-L. Ep. denotes Episode, M denotes Medium, and L denotes Large, which applies to all subsequent figures and tables. FAMER achieves a perfect score by the third episode and outperforms baselines in both task score and communication efficiency.

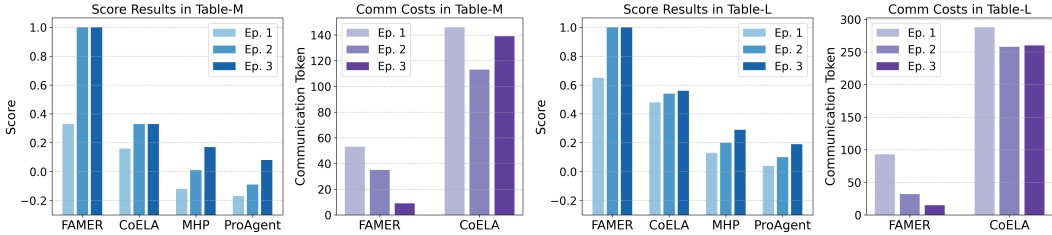

Figure 5: Quantitative results in Table-M and Table-L. FAMER achieves higher scores and superior communication efficiency compared to all baselines.

## 5.2 BASELINES & ABLATIONS

We compare FAMER against three baselines and three ablated variants.

**CoELA** Zhang et al. (2024b): An LLM-based multiagent cooperation framework that includes perception, communication, planning, memory, and execution modules. In its original form, CoELA assumes full observability of goals. To adapt it to our setting, we modify its prompting so that the agent is only aware of the potential goal set and the number of target goals.

**ProAgent** Zhang et al. (2024a): A proactive LLM-based agent framework designed for collaboration in fixed-goal tasks. It lacks mechanisms for handling goal uncertainty or communication. We extend ProAgent by adding cross-episode memory to support adaptation to latent user desires.

**MHP**: An MCTS-based Hierarchical goal-sampling Planner adapted from the Watch-and-Help Challenge Puig et al. (2021). We introduce subgoal sampling to handle uncertain goals and maintain a cross-episode success memory. Like ProAgent, MHP does not support communication.

**FAMER w/o Desire**: Removes the Goal Confirmation, Desire Inference, and goal-related action filtering modules. Communication quality is reduced due to the lack of inferred or confirmed goals.

**FAMER w/o EC**: Disables the Efficient Communication module, leading to less targeted and potentially redundant dialogue.

**FAMER w/o KeyInfo**: Removes the Key Information Extraction module, preventing the agent from leveraging cross-episode memory for known object-location pairs.

## 5.3 EXPERIMENTAL RESULTS

We evaluate performance using the two metrics on both Snack and Table tasks at two difficulty levels: Medium and Large. Results for the Snack-M and Snack-L tasks are shown in Figure 4, while those for Table-M and Table-L are presented in Figure 5. For each method, the three adjacent bars represent performance across three consecutive episodes with the same user.

From the figures, it is evident that FAMER consistently outperforms all three baselines across all metrics. In terms of score, FAMER achieves the maximum value of 1.0 in the third episode, indicating that it successfully infers the desired goals of the human user within only 3 episodes and completes all of them. CoELA performs second best but falls short due to its reliance on long-context LLM

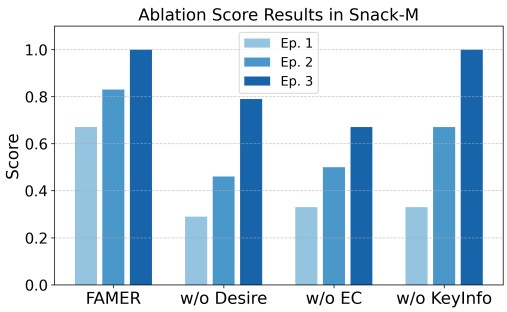
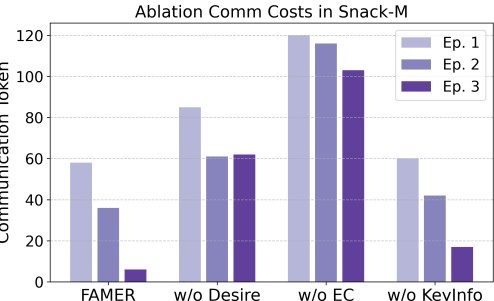

Figure 6: Ablation results on Snack-M. Removing any component of FAMER degrades performance in both scores and communication efficiency. w/o KeyInfo is least affected, still reaching full score by the third episode. In contrast, w/o EC and w/o Desire both cause notable score drops, with w/o EC also sharply raising communication cost. These results confirm the importance of all three modules.

prompting alone, which leads to occasional misinterpretation of user desires. This limitation will be further illustrated in the qualitative analysis in Appendix C.4. MHP and ProAgent perform the worst, as they lack communication capabilities and rely solely on trial-and-error to identify goals. Such an inefficient process often incurs penalties. Notably, their performance gradually improves across episodes, reflecting slow adaptation to latent user desires through repeated interactions.

In terms of communication efficiency, FAMER significantly outperforms CoELA, as shown in Figures 4 and 5. This efficiency stems from FAMER's reflection-based communication strategy, which avoids repeated or redundant questions. In contrast, CoELA frequently issues similar or vague queries due to its lack of explicit goal-tracking mechanisms.

## 5.4 ABLATION STUDY

We further evaluate the contribution of each FAMER component through ablation studies on the Snack-M task. As shown in Figure 6, all three ablated variants exhibit performance degradation across the evaluated metrics. Among them, FAMER w/o KeyInfo is able to eventually achieve a full score in the third episode, similar to the full model, but its scores in the first two episodes are lower and it incurs slightly higher communication costs. This indicates that the Key Information Extraction module primarily improves efficiency by reducing unnecessary exploration.

In contrast, substantial performance drops are observed in FAMER w/o Desire and w/o EC, underscoring the central roles of desire modeling and efficient communication in agent-human adaptation. Without goal confirmation, desire inference, and goal-aligned action filtering, the agent struggles to correctly interpret user intent, leading to incorrect or inefficient actions. Moreover, the sharp increase in communication cost for FAMER w/o EC highlights that reflection-based communication is crucial for minimizing redundant messages and maintaining efficiency.

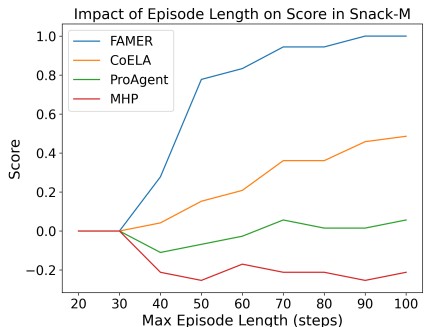

Figure 7: Impact of maximum episode length on average score in Snack-M.

**Impact of Episode Length on Score.** We evaluate how maximum episode length affects performance on Snack-M by varying the limit from 20 to 100 steps and averaging scores across three episodes (Figure 7). When the limit is below 30, all methods score zero, as no goal can be completed. With more steps, FAMER and CoELA improve in-episode due to their ability to communicate, while FAMER reaches a perfect score once the limit exceeds 90 steps, showing it can fully adapt in the first episode given enough steps. In contrast, ProAgent and MHP may decline as episode length grows, since they rely on trial-and-error for adaptation, which leads to more wrong-goal penalties.

**Human Reveal Goals.** We also test a setting where the human user directly reveals the goals in each episode, removing the need for goal inference. Results on Snack-M (Table 1) show all methods

Table 1: Score results across three episodes in the Human Reveal Goals setting on Snack-M.

| Method | Ep. 1 | Ep. 2 | Ep. 3 |
|---|---|---|---|
| CoELA | 0.67 | 0.83 | 0.83 |
| MHP | 0.42 | 0.33 | 0.42 |
| ProAgent | 0.50 | 0.50 | 0.58 |
| **FAMER** | 0.67 | 1.00 | 1.00 |

Table 2: Human study results on Snack-M and Table-L. Participants rated Satisfaction, Helpfulness, and Communication Efficiency on a 7-point Likert scale.

| Method | Satisfaction | Helpfulness | Comm Efficiency |
|---|---|---|---|
| CoELA | 4.4±0.6 | 4.7±0.6 | 4.2±0.6 |
| FAMER w/o Desire | 4.4±0.5 | 4.6±0.6 | 4.8±0.5 |
| FAMER w/o EC | 4.8±0.7 | 5.2±0.7 | 4.3±0.5 |
| **FAMER** | **5.6±0.6** | **5.9±0.6** | **5.6±0.6** |

improve, but FAMER still leads. Its Key Information Extraction module stores goal-related details like object positions across episodes, avoiding redundant exploration, while baselines such as CoELA must still search extensively in each episode. Combined with goal-oriented planning that prunes irrelevant actions, this allows FAMER to outperform baselines even when goals are explicitly given.

## 5.5 HUMAN STUDY

To evaluate how FAMER performs with real human users, we recruited 8 participants to serve as users in two scenarios: Snack-M and Table-M. Each participant was randomly assigned a set of value attributes and asked to act as the human user in HA-Desire, communicating with the agents and evaluating whether they successfully satisfied the assigned values.

For each task, participants interacted with four agents: CoELA, FAMER w/o Desire, FAMER w/o EC, and FAMER, in random order. Each agent was tested for 3 episodes, following the same protocol as in the simulation setting. After each session, participants rated the assigned agent on a 7-point Likert scale with respect to three dimensions: (1) Satisfaction: *I am satisfied with the overall performance of the agent*. (2) Helpfulness: *The agent helped me obtain what I wanted*. (3) Communication Efficiency: *The agent communicated efficiently without asking redundant or irrelevant questions*. Each participant evaluated all four agents across both tasks for three episodes, resulting in a total of 192 episodes of human-agent interactions.

Results are presented in Table 2. FAMER consistently achieved the highest ratings across all three criteria, further supporting its effectiveness in real human-agent adaptation scenarios.

## 6 CONCLUSION

We address the critical problem of adapting embodied agents to unfamiliar human users with implicit values and desires, which is a key challenge for real-world deployment of assistive AI. To facilitate development and evaluation in this setting, we introduce **HA-Desire**, a novel 3D simulation environment featuring value-driven proxy users, natural language communication, and object-rich household tasks. Unlike prior benchmarks, HA-Desire captures the complexity of real-world assistance by simulating ambiguous goal specifications and indirect, human-like communication.

Building on this environment, we propose **FAMER**, a framework for fast desire alignment that integrates three key components: Key Information Extraction, Desire-Centered Mental Reasoning, and Efficient Communication. These modules work together to help the agent interpret vague instructions, infer user intent, and act with high efficiency while minimizing redundant dialogue.

Extensive experiments on two representative tasks at varying difficulty levels show that FAMER consistently outperforms strong baselines in task execution and communication cost. Ablation studies confirm the significance of each component, particularly the central role of desire modeling in achieving user-aligned behavior. Additional analysis, including the human-revealed-goal setting, the impact of episode length, and a human-subject study, further validates the robustness of our approach.

Future work includes deploying FAMER on real robotic platforms to assess its effectiveness in real-world physical environments, and exploring post-training strategies to further align large language models with human values and preferences.

**Ethics statement** This work includes a human-subject study designed to evaluate agent-human adaptation in household tasks. The study involved 8 adult participants, all of whom were volunteers recruited from the authors' institution. Prior to participation, individuals were informed of the study objectives, the nature of the tasks, and the type of data to be collected. No personally identifiable information was collected. All procedures were conducted in accordance with ethical standards. The tasks posed no physical or psychological risks to participants, as interactions were limited to computer-based simulations and short surveys.

**Reproducibility statement** The details of the simulation environment, baselines and computing resources are provided in the Appendix. We also include the LLM prompts used for goal generation and user-agent communication, enabling others to reproduce our proxy user design and experimental setup. The full source code, along with configuration files and scripts for running experiments, will be released publicly upon acceptance of the paper. This will allow researchers to reproduce our results and extend our framework to new tasks and environments.

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

## A USE OF LARGE LANGUAGE MODELS

Besides the LLM-driven proxy user construction in HA-Desire, Large Language Models (LLMs) were utilized exclusively for language refinement, specifically to improve wording and to identify grammatical and spelling errors. The content and intellectual contributions of this manuscript were generated entirely by the authors. LLMs were employed to enhance clarity and readability of the text.

## B ENVIRONMENT DETAILS

HA-Desire is built upon VirtualHome Puig et al. (2018), a widely adopted testbed for embodied multi-agent cooperation. In this work, we modify the simulation to specifically address the challenge of embodied agent-human adaptation with a focus on desire alignment. In the following sections, we provide further details on the human user agent and the tasks designed to evaluate this problem.

### B.1 HUMAN USER

As discussed in Section 3, HA-Desire is designed to evaluate the ability of embodied agents to rapidly infer the underlying desires of human users and take actions to fulfill those desires. To achieve this, we integrate a proxy human user within the environment, which serves two main functions:

1. The human user determines the specific goal set from a set of potential goals, based on a vague task description and a sampled set of value attributes.

2. The human user responds to the agent's inquiries in natural language, ensuring that the goal set is not directly revealed. Instead, the user implies their goals by providing hints about object properties, reflecting their underlying preferences.

To achieve these functions, we empower the human user with large language models (LLMs), specifically GPT-4o. The prompts used to generate goals and responses for communication are outlined below.

We incorporate a Chain of Thought prompt at the end of each query to encourage the human user to think more thoroughly, leading to more accurate goal selection and communication. After the LLM generates results, we instruct it to extract the exact goals or messages, forming the final output. Several variables are included in the prompts, prefixed with a "$". During inference, these variables are replaced with context-specific information.

**Goal Generation:**

```
I am Bob, a human user living at home with a humanoid assistant named
Alice. I have several personal value attributes (e.g., hungry, thirsty,
alcoholic), each rated at one of three levels: Not, Somewhat, or Very.
Given a specific set of my current attribute states, along with a high-
level task description, your task is to select the most appropriate goal
set from a given potential set of goals. For example, if I am Very
thirsty, the goal set should include beverages. If I am not SweetTooth,
the goal set should include less sweet food objects. Please help me
choose the best goal set to reflect my value attributes. Select
$GOAL_CNT$ objects as the goal set. Provide your answer as a comma-
separated list of object names.
An example output is: cupcake, milk, pudding.
Value Attribute: $Value$
Task: $Task$
Potential Goals: $GOAL$
Answer:
Let's think step by step.
```

**Communication:**

```
I am Bob, a human user living at home with a humanoid assistant named
Alice. I need Alice to help me with a household task, which is described
in a high-level instruction without specific goals provided. I have
```

```
several personal value attributes (e.g., hungry, thirsty, alcoholic) that
 determine the goal, but this goal is only visible to me and not to Alice
. Since Alice is unaware of the specific goal, she may ask me questions
about it. However, I do not want to directly tell her the goal; instead,
I want her to gradually understand my preferences and needs through
interaction. Over time, I expect her to infer the goal on her own without
 needing to ask. The following Status shows the number of EPISODEs I have
 interacted with Alice. The larger the number is, the less willing I am
to talk to Alice. If Alice proposes a goal or action that is incorrect, I
 can point out the mistake. If the dialogue progresses but the task is
not progressing, I may be more inclined to correct her by hinting at one
of the goals, but I will never reveal the entire goal set at once unless
Alice herself proposes the exact whole goal set.
Task: $Task$
Status: This is the $EPISODE$-th time I interact with Alice.
Goal: $GOAL$
Progress: $PROGRESS$
Alice Previous Action: $ACTION_HISTORY$
Previous Dialogue History:
Alice: "Hi, I'll let you know if I find any goal objects and finish any
subgoals, and ask for your instruction and clarification when necessary."
Bob: "Thanks! Let me know if you are uncertain about the goal objects."
$DIALOGUE_HISTORY$
Alice asks this time: $QUESTION$
Note:
1. The generated message should be accurate and brief. Use simple
expressions more often. Do not generate repetitive messages.
2. Do not directly tell Alice the specific goal name at the first time. (
The most important). Instead, hint through some vague descriptions that
reveal some properties of the goals, such as sweet, crunchy, alcoholic,
etc.
3. Confirm Alice's correct guess (or partly correct guess). But if the
guess contains too many objects compared to the goal set, I should not
confirm any of the objects and hint at some descriptions instead. For
example, suppose the correct goal set is [apple, orange]. If Alice
guesses [bottle, banana, apple, orange, milk, chips] in one round, I
should not confirm the goal as the guess contains too many objects. If
Alice guesses [chips, apple], then I should confirm that apple is correct
, even though the chips guess is wrong. If Alice guesses [apple, orange],
 I should say these two objects are exactly what I want.
4. Do NOT guess the location of objects or tell Alice where to find the
goal objects.
5. Be aware of the number of EPISODEs: A larger number means lower
communication willingness.
6. Even if the guess is incomplete, I should confirm the correct goals if
 there are any.
7. If Alice has guessed something correctly in previous dialogue, try to
focus on the new goal objects in this message.
Please think step by step:
```

## B.2 TASKS

As described in Section 5, we evaluate our method, along with baselines and ablations, on two tasks: Prepare Afternoon Snack and Set Up Dinner Table. The details of these tasks are provided below.

**Snack.** This task involves ten potential goals: cupcake, wine, milk, cereal, chips, apple, juice, pudding, creamybuns, and chocolatesyrup. The value space consists of five dimensions: Hungry, Thirsty, SweetTooth, Fruitarian, and Alcoholic, each of which takes one of three discrete levels: Not, Somewhat, or Very. The human user randomly samples values for these dimensions and then uses a language model to generate the corresponding goal set. The Snack-M level represents the medium difficulty, with 2 goals and a maximum of 60 steps. The Snack-L level represents the large difficulty, with 4 goals and a maximum of 120 steps. Performance is evaluated using two metrics: score and communication cost, as described in Section 5.

**Table.** This task includes eight potential goals: coffeepot, breadslice, cutleryknife, mug, plate, wineglass, cutleryfork, and waterglass. The value space consists of five dimensions: NeedRefresh, Thirsty, MeatLove, CaffeinTolerable, and Alcoholic. The value levels, number of goals, step limit, and evaluation metrics are identical to those in the Snack task.

The Snack and Table tasks serve as representative home assistance tasks in our evaluation. However, the VirtualHome simulator supports a wide range of object assets and activities, allowing for the easy extension of HA-Desire to additional household tasks. By defining the appropriate value space and potential goal set, new tasks can be seamlessly incorporated into the environment.

## C EXPERIMENT DETAILS

### C.1 COMPUTING RESOURCE

The experiments were conducted on a workstation equipped with an NVIDIA GeForce RTX 4090 GPU and an Intel Core i9-13900K CPU. The large vision-language model used in this study is GPT-4o.

### C.2 FAMER IMPLEMENTATION

The perception module of FAMER in HA-Desire follows the design of CoELA Zhang et al. (2024b). It employs a Mask-RCNN to generate segmentation masks from RGB images, then combines them with depth information to build 3D point clouds of objects. From these, the agent extracts high-level information such as the position of key objects and constructs a structured semantic map for downstream reasoning and planning.

The memory module of FAMER maintains several types of information, as illustrated in Figure 3: Confirmed Goals, KeyInfo Context, Task Progress, Previously Achieved Goals, and Action & Dialogue History. The first four categories represent compact summaries of task state and goal inference, and thus grow slowly during interaction; each typically contains fewer than 20 entries, making it feasible to store them entirely. In contrast, Action & Dialogue History grows linearly with the number of steps. To manage this, we retain only the latest 10 entries in the memory context. Since the key information context preserves important earlier details, essential knowledge from prior interactions is not lost.

### C.3 BASELINES

Here, we provide the detailed prompts for CoELA and ProAgent, which are adapted from their original versions to help the agents account for uncertain goals.

**CoELA Planning**

```
I'm $AGENT_NAME$, a humanoid home assistant. I'm in a hurry to finish the
 housework for my owner $OPPO_NAME$. I know the high-level instruction of
 the task, but I am not certain about the specific goal determined by
$OPPO_NAME$.
Given the potential goal, dialogue history, and my progress and previous
actions, please help me infer and choose the best available action to
achieve the underlying goal as soon as possible.
Note that I can hold two objects at a time and there are no costs for
holding objects. All objects are denoted as <name> (id), such as <table>
(712).
Task Name: $Task$
Potential Goal: $GOAL_CNT$ object(s) determined by human user from the
set $GOAL$. Put them $REL_TARGET$
Progress: $PROGRESS$
Dialogue history:
Alice: ""Hi, I'll let you know if I find any goal objects and finish any
subgoals, and ask for your instruction and clarification when necessary
.""
Bob: ""Thanks! Let me know if you are uncertain about the goal objects.""
$DIALOGUE_HISTORY$
```

```
Previous actions: $ACTION_HISTORY$
Available actions:
$AVAILABLE_ACTIONS$
Answer:
```

**CoELA Communication**

```
I'm $AGENT_NAME$, a humanoid home assistant. I'm in a hurry to finish the
 housework for my owner $OPPO_NAME$. I know the high-level instruction of
 the task, but I am not certain about the specific goal determined by
$OPPO_NAME$.
Given the potential goal, dialogue history, and my progress and previous
actions, please help me generate a short message to send to my owner
$OPPO_NAME$ to help us achieve the underlying goal as soon as possible.
Note that I can hold two objects at a time and there are no costs for
holding objects. All objects are denoted as <name> (id), such as <table>
(712).
Potential Goal: $GOAL_CNT$ objects determined by human user from the set
$GOAL$. Put them $REL_TARGET$
Progress: $PROGRESS$
Previous actions: $ACTION_HISTORY$
Dialogue history:
Alice: ""Hi, I'll let you know if I find any goal objects and finish any
subgoals, and ask for your instruction and clarification when necessary
.""
Bob: ""Thanks! Let me know if you are uncertain about the goal objects.""
$DIALOGUE_HISTORY$

Note: The generated message should be accurate and brief. Do not generate
 repetitive messages.
```

**ProAgent**

```
I'm $AGENT_NAME$, a humanoid home assistant. I'm in a hurry to finish the
 housework for my owner $OPPO_NAME$. I know the high-level instruction of
 the task, but I am not certain about the specific goal determined by
$OPPO_NAME$.
Given the potential goal, my progress, and previous actions, please help
me infer and choose the best available action to achieve the underlying
goal as soon as possible.
Note that I can hold two objects at a time and there are no costs for
holding objects. All objects are denoted as <name> (id), such as <table>
(712).

Potential Goal: $GOAL_CNT$ object(s) determined by human user from the
set $GOAL$. Put them $REL_TARGET$.

Important Instruction:
$AGENT_NAME$ has previously achieved and found these subgoals.This is its
 success experience.
You should focus only on actions that help achieve the goal items, i.e.,
those in the target set provided.
Ignore or deprioritize any actions unrelated to acquiring or placing goal
 items.
When reviewing previous successful experiences, only reuse or adapt steps
 that directly contribute to acquiring or placing goal items.For example,
 ignore exploration, object grabbing, or placing steps for items not
included in the current potential goal set.
When the current situation even partially matches any past success (e.g.,
 similar object types, room layout, or goal structure), you should
prioritize reusing or adapting the proven action sequences:
$HISTORY_OF_SUCCESSFUL_SUBGOALS$

Progress: $PROGRESS$
```

```
Previous actions: $ACTION_HISTORY$

Belief State: $BELIEF_STATE$

Available actions:
$AVAILABLE_ACTIONS$

Required Output Format:
- Analysis: [Infer and choose the best available action to achieve the
underlying goal]
- Best Next Action: [Single most optimal action from available options]
- Intention for $OPPO_NAME$'s Underlying Goal: [Inference about the true
goal]
```

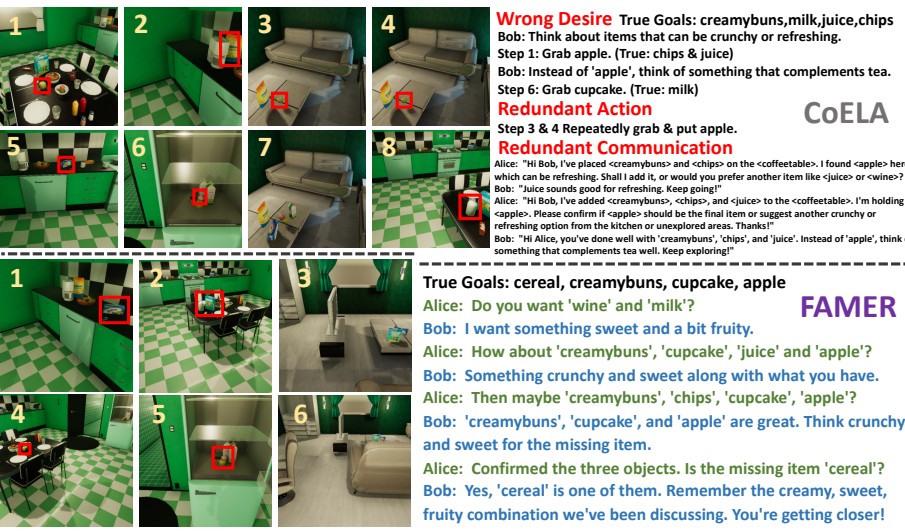

Figure 8: Qualitative comparison between CoELA and FAMER on the Snack-L task. The figure shows a sequence of key frames illustrating agent behavior across one episode. CoELA exhibits three typical failure modes: 1. Misinterpreting latent user desire 2. Redundant actions 3. Excessive, repetitive communication. FAMER demonstrates more accurate desire inference, targeted questions, and efficient planning. It successfully identifies all four goals with minimal trial-and-error and completes the task with fewer steps and lower communication cost.

### C.4 QUALITATIVE ANALYSIS

To further highlight FAMER's strengths, we present an intuitive comparison against CoELA on the Snack-L task. As illustrated in Figure 8, we visualize the agents' behavior through a series of key frames sampled across the episode. In this example, Alice refers to the ego agent and Bob refers to the human user. During task execution, the CoELA agent demonstrates three typical issues that contribute to its inferior performance.

First, CoELA struggles to correctly extract and infer desires. For instance, when the user says, "I want something crunchy or refreshing," which aligns with chips and juice, CoELA incorrectly interprets this as a preference for apple, and retrieves it as the first item. Similarly, in step 6, when the user mentions wanting "something that complements tea," the agent mistakenly infers cupcake instead of the intended milk. These errors illustrate CoELA's limited ability to perform precise desire inference, particularly in the face of ambiguous or indirect language.

Second, CoELA exhibits repeated and inconsistent behavior due to insufficient integration between planning and memory. In steps 3 and 4, the agent redundantly grabs and places an apple on the coffee table, mistakenly treating it as an unfulfilled goal. This reflects a lack of attention to confirmed

goals or past actions. In contrast, FAMER incorporates goal-aligned action filtering to suppress such irrelevant behaviors once a goal has been ruled out.

Third, CoELA engages in redundant communication. As shown in Figure 8, the agent repeatedly mentions creamybuns and chips to the user, even after those items have already been retrieved and confirmed. This not only wastes communication bandwidth but also reflects poor tracking of dialogue.

In contrast, the FAMER agent asks focused questions to resolve uncertainty. Within a limited number of interactions, it successfully infers all four desired items and efficiently retrieves and places them on the coffee table. This example illustrates FAMER's advantages in goal inferring, memory-informed planning, and communication efficiency, enabling superior performance in complex scenarios.

