# OpenReview forum: "Communication-Efficient Desire Alignment for Embodied Agent–Human Adaptation"
_ICLR.cc/2026/Conference — ICLR 2026 Conference Withdrawn Submission_

### Official Review · Reviewer_93nB · 2025-10-29

**Soundness:** 3
**Presentation:** 3
**Contribution:** 2
**Rating:** 2
**Confidence:** 4

**Summary:**

This work aims to work on human-AI interaction by designing a simulation environment and a system to allow embodied agents to infer human intentions, from a higher-level instruction, ideally a vague one, and infer concrete subgoals to execute, which can be based on a personalized set of attributes. The evaluation benchmark has been realized on VirtualHome by adding an LLM proxy of a human, which can access a pre-defined set of user value attributes, and a potential goal set. It samples some value attributes and generates a high-level goal, usually vague, that the embodied agent needs to fulfill. The proposed framework has three key components: information extraction, desire-centered mental reasoning, and efficient communication. On two tasks that are set up in the proposed simulation, their method outperforms baselines.

**Strengths:**

- The problem of working with under-specified human instructions is an important one and would eventually be helpful in enabling household robots.
- The authors’ method achieves better results on the tasks proposed, and the ablations show that each component contributes meaningfully to the final performance.

**Weaknesses:**

- I really appreciate the effort that the authors put into this work; however, I am afraid that, to me, it seems a bit difficult to see the importance of this work.
  - For instance, the benchmark adds a proxy LLM user, which is conditioned on some pre-defined set of values. I understand, it is a good way to make things more measurable and easy to experiment with, since it does not have a human bottleneck; however, in the real world, it is very unlikely that a set of pre-defined values is a good approximation.
  - Additionally, various components added are in some way preventing long-context LLM prediction, and breaking each part down into various sub-parts. This is not exactly a problem, as it is similar to the modular approach to robotics, where there are different components serving different functions.
     - But in this case, it seems that it is just making the observation processing more explicit at various places, thereby reducing one long-context pass.
    - Is the cost of repetitive querying measured somewhere?

**Questions:**

I would be looking forward to hearing the author’s opinion on the points raised in the weakness section.

---

### Official Review · Reviewer_quLo · 2025-10-29

**Soundness:** 2
**Presentation:** 2
**Contribution:** 3
**Rating:** 4
**Confidence:** 3

**Summary:**

This paper tackles the problem of fast and communication-efficient desire alignment between embodied agents and human users. The authors introduce HA-Desire, a new simulation environment that models realistic, value-driven human users through LLM-based proxies. To address the challenge of inferring users’ latent desires, the paper proposes FAMER (Fast Adaptation via MEntal Reasoning), a framework integrating three modules:
(1) Key Information Extraction for goal-relevant memory building, (2) Desire-Centered Mental Reasoning for inferring user intent, and (3) Efficient Communication for reducing redundant dialogue. Experiments conducted in HA-Desire show that FAMER improves both task success and communication efficiency compared to baseline methods.

**Strengths:**

1. Novel problem formulation: The paper explicitly frames “desire alignment” as a distinct challenge beyond goal-conditioned task completion, introducing a valuable direction for agent-human adaptation.
2. Environment contribution: The HA-Desire simulator provides a realistic and modular setup with value-driven user modeling, which can support future research in adaptive embodied interaction.

**Weaknesses:**

1. Limited experimental scope: Evaluation is restricted to **GPT-4o** as the sole underlying model. Since GPT-4o is not representative of current open or smaller models, this limits generalizability.
2. Narrow task coverage: Only two tasks (Prepare Afternoon Snack and Set Up Dinner Table) are tested, raising concerns about only adaptation in few tasks, scalability and diversity of user-goal settings.
3. Simulation-only validation: All experiments are conducted in a noise-free simulated environment. There is no discussion or testing of the framework under more realistic perception noise or physical constraints, which are critical for embodied deployment.

**Questions:**

The paper introduces an interesting and timely problem, desire alignment for embodied agents, and provides both an environment and a plausible framework to study it. However, the empirical validation is narrow (single model, two tasks, limited baselines) and needs expansion to demonstrate robustness and generality. With broader evaluation and discussion of real-world factors, this could become a strong contribution. Please refer to the weakness for more information.

---

### Official Review · Reviewer_wgcB · 2025-10-30

**Soundness:** 2
**Presentation:** 2
**Contribution:** 2
**Rating:** 4
**Confidence:** 4

**Summary:**

The paper addresses the problem of aligning embodied agents with users’ implicit desires under limited communication. It proposes FAMER, which integrates (1) Key Information memory, (2) Desire-centered mental reasoning, and (3) Efficient communication modules to reduce dialogue cost and improve adaptation speed. The authors also introduce HA-Desire, a simulated environment where a GPT-driven proxy user generates indirect hints based on human-like values. Experiments on two embodied tasks (Snack, Table) and a small-scale human study show that FAMER achieves higher success rates and lower communication cost compared to CoELA, ProAgent, and MHP baselines.

**Strengths:**

1. The paper focuses on the domain of desire alignment under communication constraints, which is a novel problem formulation.
2. The proposed HA-Desire environment adds the proxy user that may provide imprecise instructions. It is more closely aligned with reality and provide a scenario for people to train and test their models.
3. Extensive ablation and qualitative analyses (Figures 4–8, Tables 1–2) support the intuition behind each module.

**Weaknesses:**

1. Limited empirical scope and generalization. The experiments rely only on GPT-4o as the backbone model. Why not experiment on smaller open-source LLMs or even better SOTA ones. (e.g., Qwen, Deepseek / Gemini, Claude or GPT-5 series), which differ in reasoning style and grounding ability. Moreover, there is no discussion of whether FAMER’s reasoning and communication modules depend on specific model behaviors (e.g., reflection quality or instruction-following strength).

2. Narrow task coverage and lack of diversity.
(1) Only two household tasks: Prepare Afternoon Snack and Set Up Dinner Table are tested. These tasks share similar context (kitchen/dining scenes) and goal structures, limiting claims of generality to broader embodied domains (e.g., cleaning, maintenance, or navigation).
(2) The environment’s multi-room capability is underutilized; experiments seem confined to short, object-centric interactions rather than full long-horizon activities.

3. Baselines are insufficient and unclear. The paper mentions that some baselines were adapted to the task, but does not specify how these adaptations were made or whether they disadvantage the baselines (e.g., removing access to user-value priors).

4. Minor Weakness: Overreliance on simulation and lack of realism. Similar to W2, HA-Desire is a purely simulated environment built upon VirtualHome, which does not account for perception noise, action uncertainty, or physical constraints. At least, no generalization trials even based on one single model.

**Questions:**

1. How tightly is FAMER coupled to GPT-4o’s reasoning and reflection abilities? Have you tested smaller models or ablations to verify architecture-agnostic behavior?
2. You mention modifying baseline methods to fit your environment. Could you specify what those changes were and how you ensured a fair comparison?
3. Minor Question: How would FAMER perform under real-world perception noise, e.g., from camera input or sensor error? Do you foresee challenges in transferring to real robots?

---

### Official Review · Reviewer_uPxh · 2025-11-01

**Soundness:** 3
**Presentation:** 3
**Contribution:** 2
**Rating:** 2
**Confidence:** 4

**Summary:**

This paper presents a new embodied human-agent interaction environment where the human has intents that can't be explicitly revealed in communication, and the agent must infer the human's intent by looking at the actions. To tackle this challenge, the paper also introduced a modularized method FAMER to solve this challenge. Experiments show that FAMER's success rate and average communication are better than previous methods.

**Strengths:**

1. The challenge is very clearly defined and evaluation metrics are very concrete.

2. The experiments and ablation studies show better performance of FAMER against baselines and the effectiveness of the components.

**Weaknesses:**

1. Human-agent interaction where the human tries to 'hide' his/her intention do not seem practical to me. I can't think of a scenario where simply asking something like "What do you need?" won't work. That is almost surely the most communication-efficient way.

2. FAMER composes of into a goal oriented planning module and a perception-interaction module, but the goal oriented planning was not described nor experimented on.

3. FAMER's interaction module contains 'desire inference' and 'efficient communication', both do not seem to rely on the fact that the human is hiding his/her intention. It makes more sense to consider environments without the constraint.

4. Except for the planning module, all FAMER parts all fall within the scope of VLM reasoning. There should be some kind of VLM baseline where all information and requirements are sent in the prompt of a powerful VLM reasoning model instead of modularized calling.

**Questions:**

See weaknesses. Each weakness already describes some question I want to ask.

---

### Note · Authors · 2025-11-30

I have read and agree with the venue's withdrawal policy on behalf of myself and my co-authors.